# Security in Wireless Sensor Networks: A Cryptography Performance Analysis at MAC Layer

**Mauro Tropea** [1,*] **, Mattia Giovanni Spina** [1] **, Floriano De Rango** [1] **and Antonio Francesco Gentile** [2]

1    DIMES Department, University of Calabria, Via P. Bucci 39/c, 87036 Rende, Italy; mattiagiovanni.spina@dimes.unical.it (M.G.S.); derango@dimes.unical.it (F.D.R.)

2    ICAR-CNR, Via Pietro Bucci, Cubo 8/9C, 87036 Rende, Italy; antoniofrancesco.gentile@icar.cnr.it

\*    Correspondence: mtropea@dimes.unical.it; Tel.: +39-0984-494786

**Abstract:** Wireless Sensor Networks (WSNs) are networks of small devices with limited resources which are able to collect different information for a variety of purposes. Energy and security play a key role in these networks and MAC aspects are fundamental in their management. The classical security approaches are not suitable in WSNs given the limited resources of the nodes, which subsequently require lightweight cryptography mechanisms in order to achieve high security levels. In this paper, a security analysis is provided comparing BMAC and LMAC protocols, in order to determine, using AES, RSA, and elliptic curve techniques, the protocol with the best trade-off in terms of received packets and energy consumption.

**Keywords:** Wireless Sensor Networks (WSNs); BMAC; LMAC; security; symmetric cryptography; asymmetric cryptography; Elliptic Curves Cryptography (ECC); energy drain attack; impersonation attack

## 1. Introduction

In recent years, the development of hardware technologies has made it possible to create increasingly powerful and miniaturized devices. This technological advance, together with advances in wireless communications, has formed the basis for a successful new technological perspective: Wireless Sensor Networks (WSNs) [1]. The key to the success of WSNs is to be found in their versatility, in the low cost of the sensor nodes and in their highly self-reconfigurable qualities. These peculiarities have projected WSNs into various application scenarios, some of which with stringent reliability requirements such as WSNs for telemedicine and for some military applications or also smart-cities, where security is another important aspect to be taken into account [2]. Researchers are currently investigating how to manage the energy consumption of the sensor nodes more efficiently in order to increase the autonomy of the network. Moreover, how to optimally route communications from the sensor to the user, how to collect, store, and represent data with the minimum memory occupation has also attracted interest in the current literature. In particular, the need for reliable, energy-efficient data collection algorithms is a very widespread requirement as it is the basis for the monitoring and/or control of physical phenomena over a long period of time [3].

A WSN is essentially a network of devices, called sensor nodes, capable of interacting with the surrounding environment and communicating with each other in order to perform a specific task. To this end, there are three main functions that must be accomplished by the sensor nodes:

- Sensing: the measurement of physical quantities (temperature, humidity, etc.);
- Processing: the processing of the acquired measurements;
- Communication: the communication with other nodes, typically through radio frequency (RF) interfaces.

The network nodes are located within the area to be monitored, or in any case in the immediate vicinity. Usually, each node is associated with an object, a person, an animal or a decisive place for the study of the phenomenon to be observed. Typically, there is no fixed infrastructure on which nodes can be supported; for this reason they are called "ad-hoc" networks, which can be centralized if all communications are directed to a single node that processes the collected information, or distributed if the nodes have sufficient capacity and intelligence to process the data autonomously.

Figure 1 shows an example of WSN where different sensor nodes can communicate in a multi-hop manner towards the sink node (or gateway node) with the purpose of collecting data and typically transmitting them to a server or computer.

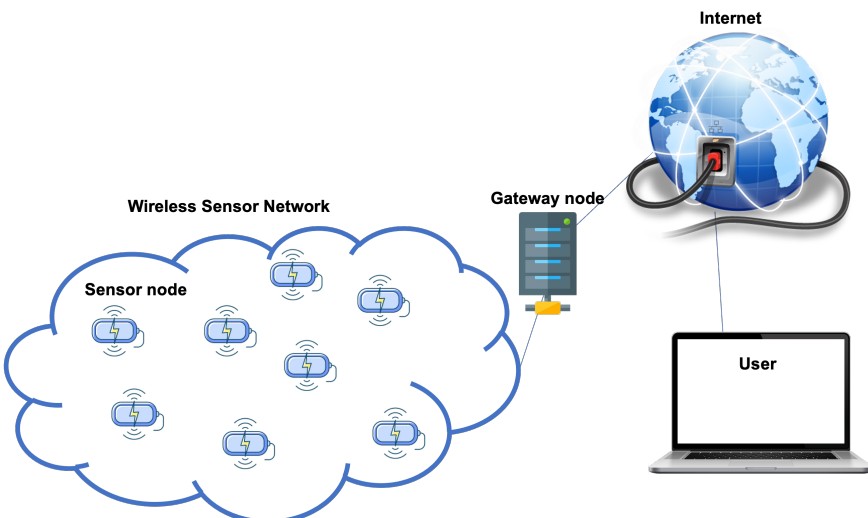

**Figure 1.** Wireless Sensor Network.

For this network typology, two aspects are very important: energy and security [4]. This is due to the particularity of sensor nodes that, unlike classic devices of local networks, are limited by battery and resource capacity (elaboration, storage, bandwidth). As a result, new medium access control (MAC) protocols that are aware of the constraints associated with each individual sensor in the network are required [5,6].

WSNs have a wide range of applications; however, as the sensor network becomes more complex, the security of WSNs becomes even more important. The main security threats are due to the location of sensors in remote areas and their geographic distribution. Unfortunately, traditional safety mechanisms are not usable for sensors, as they introduce high overload. Several researchers have provided different studies on efficient security mechanisms that respect, in fact, the constraints in terms of resources and memory imposed by WSNs [7].

Sensor networks need basic security services, like any other communication system, in order to protect data and resources from any potential attacks.

Sensor networks need light encryption to achieve a high level of security. Each sensor should have a balance between cost, performance and level of safety; however, it is very difficult to achieve these goals at the same time. In some circumstances, developers sacrifice the level of security by using cost-effective solutions without adequate mechanisms, for example, for key distribution. A WSN requires more flexible methods for distributing keys across the network. There are two types of approaches: classical, based on the use of symmetric or asymmetric cryptography, and advanced, based on elliptic curve cryptography (ECC), they are particularly known for a more efficient use of resources than any other public key technique.

WSNs should send data in a safe way towards a central node in a multi-hop transmission guaranteeing data confidentiality and integrity.

There are numerous attacks and threats toward a WSN, for example, information loss, communication interruption, service slowdown causing delays, denial of service [8,9]). Thus, we have taken into account three of the main security properties: confidentiality, integrity, and authentication. Researchers propose different countermeasures such as intrusion prevention, intrusion detection [10,11], and cryptography approaches. In this paper, starting from a previous work [12] where we have shown a MAC protocols comparison for that concern received packets and energy consumption of the sensor nodes inside the WSN, we will show an analysis on the security aspects for WSNs based on cryptography mechanisms considering energy drain and impersonation attacks and their possible mitigation.

This paper has the following structure: Section 2 presents related work on energy and security aspects in WSNs; Section 3 provides an analysis of the cryptographic methods used for experimentation; implemented attacks with relative countermeasures are provided in Section 4; Section 5 shows the results of the performed experiments; lastly, Section 6 summarizes the work.

## 2. Related Work

The design of a good MAC protocol for WSNs is a hot and important topic to be taken into account. The energy consumption is critical because the nodes that compose these networks are normally fed by batteries, and so it is important to guarantee them high autonomy and throughput both in the transmission phase and in the data elaboration [5]. Another important aspect related to energy issues concerns security that in this network typology can particularly affect the nodes due to their limited resources [13].

### 2.1. MAC Energy Issues on WSNs

A network's energy efficiency may be improved at each protocol stack layer, and the MAC layer represents a critical point in WSN energy issues [14]. In the literature, various MAC protocols have been developed, and many of them deal with energy aspects [6].

Many methods are proposed with the main task of improving the energy efficiency of these systems at MAC layer. An approach based on MAC for enhancing energy efficiency is provided in [15]. The solution provided by authors incorporates energy harvesting devices (EHDs) and analytic information, such as the estimation of the network lifetime. Moreover they provide a review of the MAC protocols that deal with this topic.

For a WSN, power drainage is a priority issue associated with the MAC layer. So, in [16] the authors propose an improvement of the sensor medium access control (SMAC) operating on its fixed duty cycles. Their idea is to break duty cycles into micro ones with a variable time mechanism. They prove the best performance of the proposal.

In [17] the authors compare two MAC contention-based protocols in terms of energy: SMAC and TMAC. Nodes using these protocols can be in sleeping or listening state. Nodes are able to save energy while they are in the sleep state by turning off the radio module. Their analysis is proved by simulative results.

Duty cycling is chosen as the major mechanism for conserving energy in WSNs. Many studies deal with this mechanism in order to analyse the efficiency of MAC protocols. In [18] the effect of duty cycling in MAC protocols and their effect on energy consumption is analysed by the authors throughout the use of a tunable MAC protocol able to perform tuning of the duty cycle to see the energy levels.

In [19], a proposal is made to reduce the problem of idle listening or collisions, as well as the energy consumption of the existing S-MAC protocol. With a study of the trade-off between energy consumption and delay, the influence of duty cycle change in line with data traffic has been given throughout the use of NS2 software simulator.

### 2.2. MAC Security Issues on WSNs

The state of the art proposes different approaches to secure communication in WSNs [20], and it shows that the cryptography is one of the common solutions for providing security.

Some works present an overview on the security issues over WSNs showing attacks, vulnerabilities and potential countermeasures [21,22]. They show that, generally, the security aspects in WSNs try to resolve issues due to four main concepts: confidentiality, integrity, authenticity, and availability [23]. Clearly, for the nature of WSNs, the nodes inside these networks have to provide more effort than in the other networks for their characteristics of elaboration, memory, and storage capacity [21]. Moreover, a deep analysis of the main protocols is needed for being able to quantify the effort to deal with security at the different protocol layers [22].

A deep study on MAC protocols concerning security attacks is in [20]. In this paper, the authors have provided an analysis on different MAC protocols discussing their security issues. They, on the basis of the detailed review, indicate the advantages and disadvantages of the studied MAC protocols along with countermeasures for the security issues.

Sensor nodes under attack might effect the accuracy and integrity of information by misrepresenting sensed data in the process of aggregation in order to send information towards the sink node. In [24], in order to deal with these security concerns, the authors provide the study on a novel approach for WSNs MAC safe data aggregation based on homomorphic propriety and called HMSDA. This approach has been shown to offer acceptable data confidentiality and integrity protection in a WSN.

A novel cryptography approach called aggregating secure data-separate MAC (SDA-SM) is shown in [25]. The proposed mechanism is able to separate the protected aggregate data and the message authentication codes into two distinct messages and, further, transmit these messages in a new random time-slot as indicated into the TDMA scheduling. The authors show that SDA-SM reaches its goal respecting a lesser computation time in the communication overheads than previous approaches. Furthermore, SDA-SM is proved to maintain the correct security proprieties of aggregate data, avoiding energy waste to prolong the lifetime of the network.

The proposal of a modified version of the Z-MAC protocol with the introduction of the elliptic-curve encryption (ECC) technique is provided in [26]. The authors analyse the MAC protocol able to transfer data in a safe manner. This new protocol is proved capable of offering low contention, high throughput, reduced latency, low power consumption, and increased efficiency.

### 2.3. Main Contributions of the Paper

Since WSN networks are highly vulnerable, due to the wireless communication channel and the extremely low resources, their security aspects represent an important field of research. Many studies analyze the behavior of different MAC protocols in the context of security. In our previous contribution [12], we have proposed a comparison between two different MAC protocols belonging to different categories, scheduling based (TDMA) or contention based (CSMA): Lightweight MAC (LMAC) [27,28] and Berkeley MAC (BMAC) [29,30] respectively. We have provided some details on both protocol operations showing the performance evaluation and results on the basis of simulation performed in a well-know software simulator: OMNeT++. Throughout the tests, we have proved the greater efficiency of LMAC protocol in respect of BMAC in terms of received packets and energy consumption. Starting from this contribution [12] we continue the study of MAC protocols also regarding security aspects.

Unfortunately, traditional safety mechanisms are not usable for sensors, as they introduce high overload. Several researchers have proposed various optimized security schemes that respect, in fact, the constraints in terms of resources and memory imposed by the WSNs. So, they need light encryption to achieve a high level of security.

In this paper we present encryption approaches based on key distribution using advanced encryption standard (AES), rivest shamir adleman (RSA) [31] and elliptic curve cryptography (ECC) [32] techniques. Then, we show a comparison between these three approaches in a WSN using BMAC and LMAC protocols considering two different types of attacks: energy drain and impersonation.

## 3. Analyzed Cryptographic Methods and Primitives

WSNs, as we said, paved the way for a wide range of possible useful use cases involving, as a consequence, several kinds applications such as the ones related to the military field or the scientific field but, also, some applications related to smart houses and home automation. But, considering these critical-task use cases, there is a growing interest in security issues which could be related to these fields. Due to the power and energy resources constraints which characterize the devices which are involved in these kinds of networks, it is fundamental to find an adequate security mechanism which takes into account all these constraints and tries to optimize the resource usage minimizing the resource wastage. So, as a consequence, it is not possible to use the traditional security protocols and schemes, since these are considered unsuitable with respect to the design requirements of the sensors and, therefore, a more careful study and analysis is needed. Thus, in this section, the security mechanisms and schemes considered in our analysis are widely described along with the guaranteed security properties.

### 3.1. Advanced Encryption Standard (AES)

The AES is a symmetric block cipher developed as a replacement of the Data Encryption Standard (DES) algorithm which has been broken in the seventies. AES is a standardized and widely used cipher which allows the use of three different key lengths: 128 bits, 192 bits, and 256 bits. It was first named "Rijndael", name which derives from the two belgian cryptographers who developed it. In 2002, it was re-named the Advanced Encryption Standard and it was published by the NIST (National Institute of Standards and Technology). It was, also, approved by the National Security Agency (NSA) institute in order to handle top secret information and it has become a standard cryptographic mechanism since then. Nowadays, AES is very widespread. Plenty of libraries have been developed in several programming languages such as C, C++, Java, and Python. Even WhatsApp and Facebook, today, use AES in order to make the exchanged messages confidential. Furthermore, it is so standardized that there are several hardware implementations for AES operations in Intel and AMD processors.

### 3.1.1. Confidentiality in AES

Due to the resource constraints which characterize WSNs and the involved devices, we decided to use a key length of 128 bit for AES. In order to make exchanged data confidential, which means that only sender and receiver can understand the data carried within the packets. The packet field which we want to make confidential is ciphered using AES-128 considering the symmetric shared key between the server and the sensor. To gain a better security degree, during the ciphering operation we randomly use an Initialization Vector (IV) since the first block of the plaintext, before being ciphered, is involved in an XOR operation with the IV, so the IV cannot be predictable and must be as random as possible. So, sensor nodes will send to the server a packet which contains a ciphered field which is obtained applying the AES to the field we want to make confidential with the shared key between sensor and server.

The entire security of the communication described in Figure 2 is based on the assumption that only sender and receiver can access and know the symmetric shared key. As an additional operation, in order to avoid special characters in the packet which could lead to packet loss during the transmission, we encode, using the base64 method, the ciphered field.

### 3.1.2. Integrity and Authentication in AES

To achieve integrity and authentication for the exchanged packets we use a tag. To this purpose, we use the cipher-based message authentication code (CMAC) algorithm. Following NIST specifications, CMAC is equivalent to one-key MAC 1 (OMAC1): it is an authentication code for messages and it is based on block cipher algorithms. The initial step involved in CMAC is, first, to choose a block cipher *E* (and we used AES) and then the

length of the CMAC code *t*. So, before sending a packet, it is generated, using AES and the shared symmetric key, a tag. Then, when the server receives a packet, it will accept it only if the tag-verify step returns "true", meaning that the packet has not been modified while transmitting it. Instead, if the tag-verify step returns "false" it means that the received packet has been illegally modified. Thus, while ensuring data integrity, using this method we ensure that data are generated only by entities which are part of the networks, i.e., they share a valid symmetric key. All the operations made in order to achieve confidentiality, integrity and authentication are shown in the Figure 3.

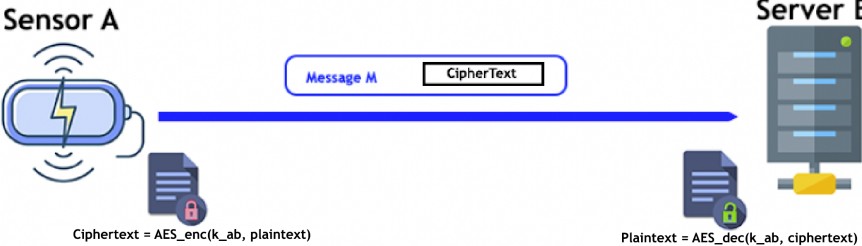

**Figure 2.** AES sent packet.

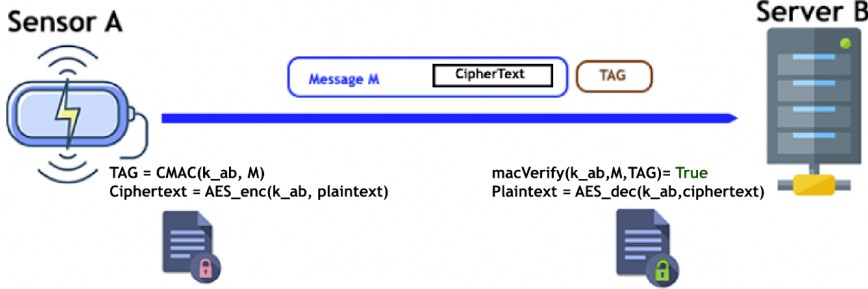

**Figure 3.** Confidentiality, Integrity and Authentication in AES.

### 3.2. Rivest Shamir Adelman (RSA)

The RSA is a cryptographic algorithm which belongs to the asymmetric (or public key) algorithm. Public key cryptography is an algorithmic, cryptographic system that is based on the use of two keys, a public and a private key used to cipher and decipher data. Particularly, if we use one of the keys to cipher the data, for example the public one, we must use the other, the private one, in order to decipher the data and get the right plaintext. Despite the slight relationship that exists between these two keys, all the public key cryptography security is based on the assumption that it must be impossible to retrieve any kind of information about one of the two keys starting from the other. These kinds of algorithms are based on mathematical relationships that have no efficient solution. These relationships make it extremely difficult for anyone to get the private key based only on the public key, which can be, for example, sniffed with a man-in-the-middle attack during the communication between the two legacy entities; and this is the foundation of the security of the public key cryptography. The security strength behind the RSA algorithm, indeed, is built upon two specific mathematical problems: the problem of doing factorization of large numbers, which is computationally infeasible when the involved numbers are too large; and the RSA problem which states that computing the private key when given only the public key must be an operation which cannot be performed in polynomial time, so it is a computationally infeasible operation too. Using RSA, in order to guarantee an adequate security, it is necessary to use a key with a size of, at least, 2048 bits. Keys of size 512 bits are breakable in a few hours. While, keys with a size of 1024 bits are widely used yet but it is preferable not to use them. Indeed, nowadays, due to the increasing power capability of dedicated hardware, it is possible to factorize an integer of 1024 bits in just one year but with a cost of one million of dollars. So, when using such a key length it is preferable to change the keys over time to get a higher degree of security. RSA is able to manage

message of limited size. Using keys with a higher size make it possible to cipher messages with higher size.

### 3.2.1. Confidentiality in RSA

We ensure confidentiality with RSA by ciphering the message with the public key of the target server, which upon the receipt of the message will decipher it using its own private key see Figure 4. This is the traditional schema used to ensure confidentiality using the public key cryptography. This will guarantee that only those who know the private key, i.e., the server entity, will be able to decipher the message. Of course, the security of public key cryptography is based on the fact that no one except the possessor of the private key is able to get information about this key.

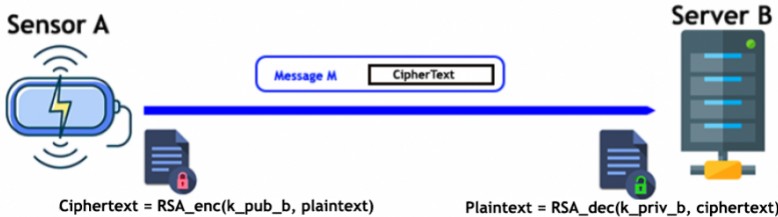

**Figure 4.** RSA sent packet.

### 3.2.2. Integrity and Authentication in RSA

To achieve integrity and authentication of the exchanged data we leveraged the digital signature technique. Each sensor will be featured by a couple of keys, a public and a private one. For the case of integrity and authentication alone, without considering confidentiality, packets, being sent by each sensor to the server entity are composed of the plaintext message along with the digital signature computed upon the message using the private key of the sender, ensuring, in this way, that no one can impersonate him/her (unless their private key is leaked, but we have to always suppose that). On the other hand, when the server receives the message along with the digital signature, it has to verify this signature. In order to do that, it has to apply the RSA algorithm to the tag, in order to retrieve the original content of the tag itself, and then compare it with the plaintext received. If the messages are the same it means that no one has modified it during its transmission over the channel. Considering, now, all the security properties apply at once as shown in the Figure 5.

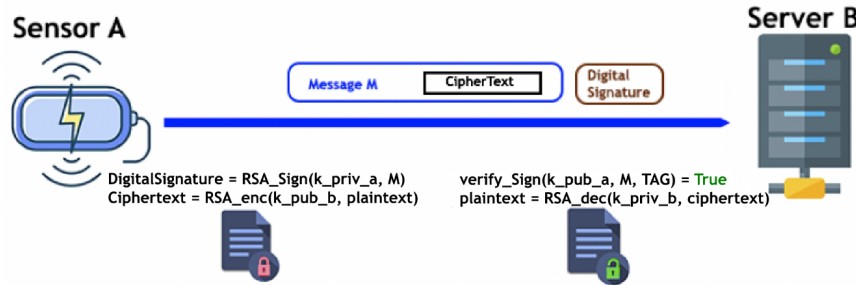

**Figure 5.** Confidentiality, Integrity and Authentication in RSA.

### 3.3. Elliptic Curve Criptography (ECC)

The ECC is a type of cryptography which is encompassed in the public-key cryptography class. Its functioning leverages on the algebraic structure of the elliptic curves defined over finite fields. It could be used for several tasks related to the field of cyber security, but the two more important are secure key exchange (with the Elliptic Curve Diffie–Hellman Algorithm, also known as ECDH), digital signature (with the digital signature algorithm (DSA) extended to the use of the elliptic curve, which is called ECDSA). Many works propose to use ECC in the Internet of Things (IoTs) field, for example making the MQTT

protocol more secure by mitigating data tampering, eavesdropping and replay attacks such as [33,34]. The ECC algorithm was developed after RSA due to advancements in terms of computational power and more advanced factorization techniques. These advancements, as a consequence, determine the use of higher key sizes and, thus, make the ciphering/deciphering operation as well as the generation mechanism of the keys more resource-hungry. The security of the ECC is based on the infeasible discrete logarithmic problem (DLP) [35], which by now is an unsolved problem, not solvable in a polynomial time. And this problem is also more difficult to solve when extended to the ECC field, the so called ECDLP [36]. Therefore, as a consequence, considering the same security degree, this kind of cryptography needs public keys of lesser size, and thus it is more suitable to this constrained context than the RSA. The NIST stated that a key of 256 bits used in ECC, for example, can guarantee the same security of 3072 bits used by RSA.

### 3.3.1. Confidentiality in ECC

As a traditional public key cryptographic algorithm, the ciphering and deciphering steps are carried out as we described when describing RSA related to confidentiality. The only difference in the case of ECC is the key generation mechanism which is based on the geometric element called elliptic curve and it has been proved that this mechanism is very efficient and optimized. So, the sensor uses the public key, generated using ECC, of the server (the target) to cipher messages, and then the server upon the receipt of the messages uses its own private key, also generated using ECC, to decipher the ciphertext and get the plaintext, see Figure 6.

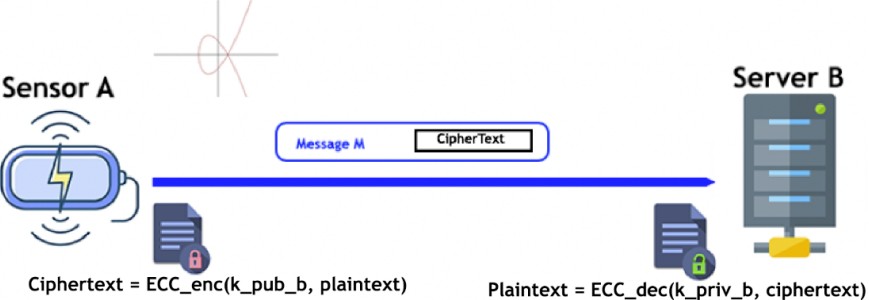

**Figure 6.** ECC sent packet.

### 3.3.2. Integrity and Authentication in ECC and ECDSA

Digital signatures [37,38] generated using ECDSA guarantee authentication and integrity. The procedure done to generate and verify the digital signature is the same of that used for the RSA algorithm. So, the node will generate the digital signature using its own private key and then the server, upon receipt of the tag, in order to verify that the messages are not modified illegally during the transmission, will check the validity of the tag using the public key of the sensor. The whole security mechanism, ensuring confidentiality, integrity and authentication is depicted in Figure 7.

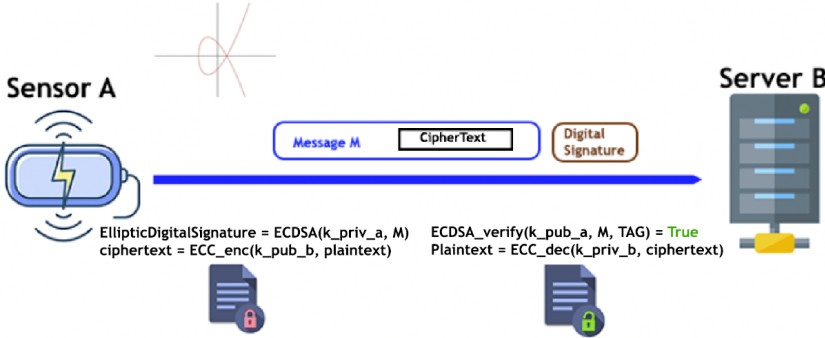

**Figure 7.** Confidentiality, Integrity and Authentication in ECC.

## 4. Implemented Attacks and Relative Mitigations

A WSN should always be able to resist attacks so that it can continue to provide its services and, therefore, it is also necessary that mitigations are introduced for allowing it to continue a normal network operations [39].

The attacks that will be proposed in this section are:

- energy drain attack;
- impersonation attack.

### 4.1. Energy Drain Attack

As said before, WSNs, in the IoT field, despite the huge benefits they can provide, raise not negligible security issues due to the inherent security vulnerabilities which could be found inside the communication protocol or the IoT devices which are involved in these particular networks. Leveraging these vulnerabilities, an attacker can focus its malicious intent on the most resource- and power-constrained IoT devices: sensors and actuators which are part of a WSN. The idea, related to the energy drain attack, also named energy depletion attack [40] is the following: the attacker aims to overwhelm the target device, in order to force it to execute energy-hungry operations such as receiving a large number of ping messages, for example. This attack can lead to the complete draining of the devices' batteries very rapidly. Extending this kind of attack to all the sensors and actuators composing the WSN, it is clear that to successfully target each device and complete this attack will lead to the shutdown of the whole WSN. Most of the existing energy drain attacks target the MAC layer. The traditional strategy, adopted by an attacker, is: the attacker makes a broadcast of forged packets including in them simply garbage data which have, however, to be verified and processed by the receiver device, consuming its energy. This kind of attack is also named as garbage data verification, since the receiver device, in behalf of the attacker, has to verify data which are essentially forged garbage. Another name which can be used to refer to this kind of attack is denial of sleep or sleep deprivation torture [41]. The denial of sleep, applied to powered devices, reflects in the energy depletion. This is an attack that could be made using several exploits such as executing collision attack, repeating handshaking operations, flooding the node with useless ping messages and so on, with the common aim of preventing the node from switching into its sleep phase.

#### 4.1.1. Energy Drain Attack Targeting a Specific Sensor

As previously stated, the attack aims to drain the energy of a specific sensor node which is part of the WSN unless its energy supplies are entirely consumed and, in this way, cause a denial of service. In our scenario, the attacker sensor node which can gain access to the WSN floods the target sensor node with a huge amount of packets. In order to make it possible, the "send_interval" parameter of the attacker node must be smaller than the one of the legacy sensors. Of course, this parameter can be set inside the environment we used to simulate the scenario, which will be described in the next sections. Reducing the value of this parameter, for the attacker sensor node, this sensor node will be able to send packets with a higher frequency and will easily overwhelm legacy sensors of the WSN. Particularly, if the "send_interval" parameter for the sensor node is set to 1 s, for the attacker we choose a reduced value of 0.1 s to make the attacker able to send a packet each 100 ms. Of course, the attacker sensor node itself, performing this kind of attack, would rapidly drain its energy supply. Thus, we consider, without loss of generality, an ideal energy model for the attacker where simply: the resources of the attacker sensor are infinite. This is not so far from the reality, since normally, the attacker must have more capacity, both in terms of power and energy, than the attacked devices.

#### 4.1.2. Energy Drain Attack: Mitigation

In order to mitigate the energy drain attack, the attacker should have the same behavior of legacy nodes of the WSN, and then the same legacy node send_interval value. In practice, to avoid this attack, it is necessary to identify a non-legacy interval time to determine if there

is a malicious node carrying out packet flooding. If a legacy node detects a send_interval smaller than the fixed legacy one, it stops the attacker node for a fixed time interval. This allows it to reduce and limit the number of packets which the attacker can send on the network and, therefore, to prevent the energy draining of the target sensor node. In this way, the whole network will work as normal.

*4.2. Impersonation Attack*

The impersonation attack is meant to make the attacker sensor node appear to be a legacy node of the WSN, violating the authentication property. It is represented by a malicious node with a malicious behavior sometime known as selfish attacker. There exist several forms of the impersonation attack [42,43]. The first is the *invisible node attack* in which a malicious node $A$ is posed among two legacy nodes $L1$ and $L2$ which are not in direct range. The attacker sensor node $A$ replicates the communication between $L1$ and $L2$, which will assume, wrongly, that they are directly communicating. In this way, the malicious node $A$ is able to impersonate the node $L1$ with respect to $L2$ and the node $L2$ with respect to $L1$. The other type of impersonation attack is the *stolen identity* attack. The malicious node $A$ is able to steal all the authentication credentials from a legacy node $L$, such as the public key used for the DSA. If the malicious node $A$ can update these credentials before the legacy node can do it, these credentials, of the legacy node $L$, will not be valid anymore. Therefore, only the malicious node $A$ will be able to communicate with the node $L$. In this case, the attack is not limited only to stealing the identity of a node, but it is also something that is related with abusing trustworthiness relationships that others sensors/entities may have established in a previous moment with the legacy node $L$.

4.2.1. Impersonation Attack: Impersonating a Sensor Node against the Legacy Server

In our scenario, the attacker, before carrying out the attack and just before sending the packet to the server, modifies the source IP address in the packet so that the other node thinks that it is a legacy node of the WSN. This modification, made by the attacker, is done at network layer exploiting the IPv4 protocol. Particularly, the attacker forges a new packet and in the decapsulation phase, sets this packet as a source IP one that is assigned to a legacy node in the WSN, choosing it in a random way. The packet, therefore, will be forged by the attacker but will traverse the network with a source IP address which cannot be referred in any way to the malicious sensor. Considering a scenario which does not provide any cryptographic technique, there are no mechanisms which could guarantee the proper sender identification; thus, the attack is completed and the server will be fooled regarding the real identity of the sender.

4.2.2. Impersonation Attack: Proposed Mitigation

It is more complex to mitigate this kind of security issue. So, we propose several techniques with relation to the specific cryptographic method that is used. The techniques we will consider are the following ones:

1. Symmetric cryptography;
2. Public key cryptography.

4.2.3. Use of Symmetric Cryptography

Considering the scenario in which symmetric cryptography is used, each sensor shares with the server a secret key, which is supposedly unknown to anyone except by the two entities, the sensor, and the server. In order to mitigate the attack, therefore, sensor nodes send to the server their packets, which are composed of the message $m$ (as a plaintext), a *ciphered field* (the one we want to make confidential) and a *tag*.

Upon the receipt of the message, the server, first of all, will verify the authenticity of the source verifying the tag. If the tag-verify operation is successful, then it proceeds with the deciphering operation. Suppose that, considering the scenario described above, there exists an attack attempt, the attacker sensor node forges a new packet with the message

*m* (as a plaintext) and a *tag*, generated using the key shared with the server. Also, in this case, the attacker will change, as we explained earlier, the source IP address of the packet impersonating a legacy sensor of the WSN.

Then, upon the receipt of the packet, the server, considering the packet as generated by a legacy sensor, will go through the verification phase of the tag using the key shared with the legacy node, described by its source address. Of course, the verification will fail since the *tag* provided to the server is generated using the key of the malicious node and not using the key of the legacy node. Thus, the server will be able to immediately detect the attack attempt, and refuse to decipher the data.

### 4.2.4. Use of Asymmetric Cryptography

In this case, each sensor node will keep a couple of keys: a public key and a private key. A packet, sent by a node, will be composed of the message *m* (as a plaintext) along with the digital signature, obtained from the plaintext, and the private key of the sender, see Figure 8.

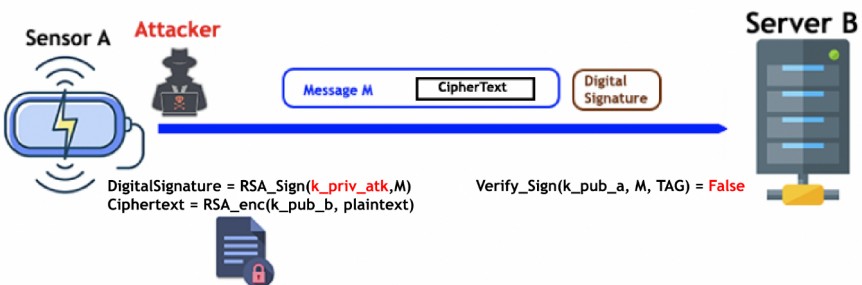

**Figure 8.** Assessing integrity using digital signature considering an attacker.

As we explained in the previous sections, the server will use the public key of the sender to verify the signature. Also, in this case, an attack attempt has been considered. As the previous example related to the symmetric cryptography, also here the attack will fail since the signature verification will fail due to the signature generated not with the private key of the actual legacy sensor but with the private key of the attacker node. Thus, when the server will retrieve the public key of the legacy sensor in order to use it to verify the tag, the verification process will fail since the key is not the right one. Of course, regarding asymmetric cryptography, either using RSA or ECC, the aforementioned process is the same, but in the latter case, using ECC, the only different procedure is in regard to the generation of keys: a procedure that involves the use of the geometric theory and the related geometric element called elliptic curve.

### 5. Performance Evaluation

This section provides details about the considered experiments performed using a software simulator. We will show the network configuration aspects providing information about the number of nodes used for simulations. In Figure 9 the main parameters of wireless interface are shown. Moreover, we provide information about the BMAC and LMAC settings (for details on these two MAC protocols refer to [12]), and then we show the results about the two types of considered attacks: energy drain and impersonation attacks. Lastly, a brief comparison between BMAC and LMAC is provided in order to show the performance of the WSN in terms of energy consumption and total number of received packets.

```
# radio and radioMedium
**.radio.centerFrequency = 2.45GHz
**.radio.bandwidth = 2.8MHz

**.radio.transmitter.bitrate = 19200 bps
**.radio.transmitter.headerLength = 8b
**.radio.transmitter.preambleDuration = 0.0001s
**.radio.transmitter.power = 2.24mW

**.radio.receiver.energyDetection = -90dBm
**.radio.receiver.sensitivity = -100dBm
**.radio.receiver.snirThreshold = -8dB
```

**Figure 9.** Wireless Interface parameters.

### 5.1. Experimental Setup

We used OMNeT++ [44] in order to set up the entire simulated WSN, both to simulate the normal behaviour of the network, and also the behaviour of the network under the attacks described in the previous sections. We used the version 5.6.2 of OMNeT++, the last version at the time we made the experiment. To support OMNeT++, we added the INET framework [45] which provides several simulation models for WSN suited to our specific study. For example, it contains models to emulate the Internet Stack (TCP, UDP, IPv4, OSPF, BGP, etc. . . ), and data link protocols such as Ethernet, PPP, IEEE 802.11, several MAC protocols suited to sensors and other interesting features. We used INET version 4.2.0.

### 5.2. Network Configuration: Sensors, Gateway and Server

In order to provide more effectiveness and significant experimental comparison and results, we developed two kinds of WSNs, which differ only for the number of deployed sensors. Before describing how networks are composed, we have to mention that in the "omnetpp.ini" we set the "send_interval" parameter to 1 s, so each node could send one packet per second. Considering the two developed networks, we have:

- A network composed of 4 sensors, a gateway and a server: in this setting, each sensor will send 100 packets during a simulation time of 100 s. So, considering four sensors, 400 packets will be sent.
- A network composed of 8 sensors, a gateway and a server: also in this case, in 100 s of simulation time, each sensor sent 100 packets so 800 packets will be sent.

For simplicity, we show a picture which describes the network with four sensors:

To better understand how the overall communication works, each sensor node transmits, in a periodic manner, data towards a gateway node using a wireless channel, which in turns, transmits this data to a server using a cabled connection. In the Figure 10, an ideal warehouse blueprint in which all the described elements are placed in is depicted. As we can see from the picture, the network is characterized by a star topology. The wireless interface of sensor nodes and gateway are specified, also, in the "omnetpp.ini" and, all the specifications, used to set all the wireless interface parameters, are listed in the Figure 9.

Each sensor node sends a UDP packet with a fixed payload size of 10 bytes, each second, to the server. Packets will be characterized by a fixed UDP header of 8 bytes and an IPv4 header of 20 bytes so, for the MAC layer a frame header of 38 bytes has been considered. Each packet sent by a sensor to the server will be routed by the gateway.

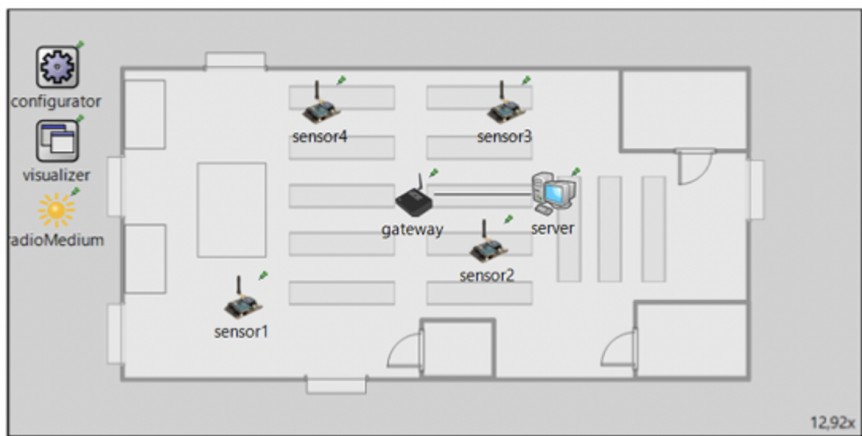

**Figure 10.** Sensors and server deployment inside the environment.

*5.3. Network Configuration: BMAC and LMAC Setting*

We made a preliminary and in-depth study in order to optimize the total packets number transmitted over the network considering the optimization of the slot duration. In this work, all studies and simulations made for the optimization of this parameter are not shown. We focus only on the security issues over WSNs indicating the final choices. In particular, we set for the BMAC protocol a value of the slot duration equal to 0.025 s, and a value of 0.050 s for the LMAC one, such as references in [12].

*5.4. Performance Evaluation: Energy Drain Attack*

We consider the attack scenario taking into account the two protocols: BMAC and LMAC. It is important remember that with LMAC protocol the attack does not fully drain the energy of the target sensor, since LMAC handles the energy consumption in an optimized way. Moreover, we have considered as power model for the nodes the *idealEpEnergyStorage*, which is an ideal power model with infinite energy and power supply. But this is not so far from the reality, since the attacker sensor has to be more powerful than the legacy sensors in order to be able to completely perform the attack. In order to carry out the performance evaluation, considering the energy drain attack, three parameters are considered:

- Total energy consumption over the total number of received packets;
- Total number of received packets;
- Number of packets lost.

The target sensor is sensor1. In particular, the attacker node will hijack and re-route its traffic to sensor1, which will not be able to handle it entirely, and even before the simulation could end (specifically after 49 s) its energy will be completely drained, causing, therefore, its inability to contribute in the WSN. This situation is depicted in Figure 11.

In order to get a more fine-grained comparison, both scenarios, the one not under attack and the other one with the energy drain attack, are considered. Each node features an energy supply of 1 J (Joule), while the attacker node, as said, has infinite energy resources. The experimental results are shown in Figure 12.

As it is possible to see from Figure 12, if the attack takes place, the energy consumption is almost doubled compared to the not under attack scenario. Considering the total number of received packets, a slight reduction is noticeable, starting from 155 to 123 packets, due to earlier energy draining of sensor1 which goes offline before the simulation can complete, and therefore it is not able to send all 100 packets. The number of lost packets, instead, is high since the attacker sends all its packets to the sensor1, and thus, they will never be received by the server.

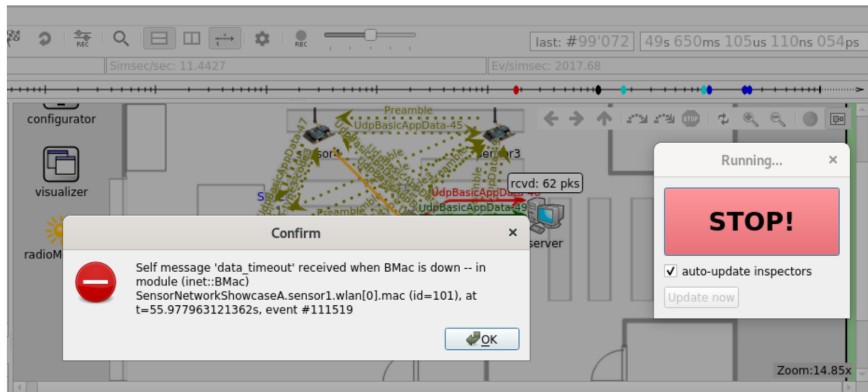

**Figure 11.** Simulation with drain attack towards sensor1.

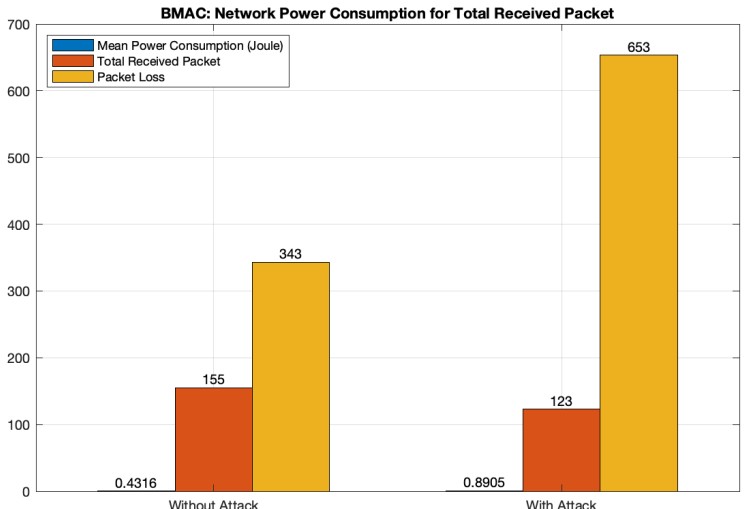

**Figure 12.** Average energy consumed, total number of received and loss packets for BMAC with and without energy drain attack.

### 5.5. Performance Evaluation: Impersonation Attack

Also for the impersonation attack, the BMAC and LMAC protocols along with the same evaluation parameters considered in the previous attack will be described.

These parameters have been evaluated considering different cryptographic methods: AES, RSA, and ECC. And, since the attacker sensor node is placed inside the network, it is possible to observe higher traffic and, as a consequence, a higher energy consumption.

#### 5.5.1. Impersonation Attack with BMAC

Considering AES and ECC, for each sensor, an energy supply of 1 J and a slot duration of 0.025 shave been used. Instead, for RSA, an energy supply of 2 J and a slot duration of 0.060 shave been set for each node. These choices are due to RSA using keys with higher size in order to guarantee a correct system behavior and to prevent sensors' energy drain. The experimental results are shown in Figure 13.

As we can see from the Figure 13, considering the attacker sensor node inside the network and applying no mitigation, the total number of received packets is 155, and the energy consumed is 0.431655 J with packet of 10 bytes. The total lost packets, instead, is equal to 343. Introducing the use of cryptography, to mitigate the attack, network performance will show notable variations with respect to this initial setup.

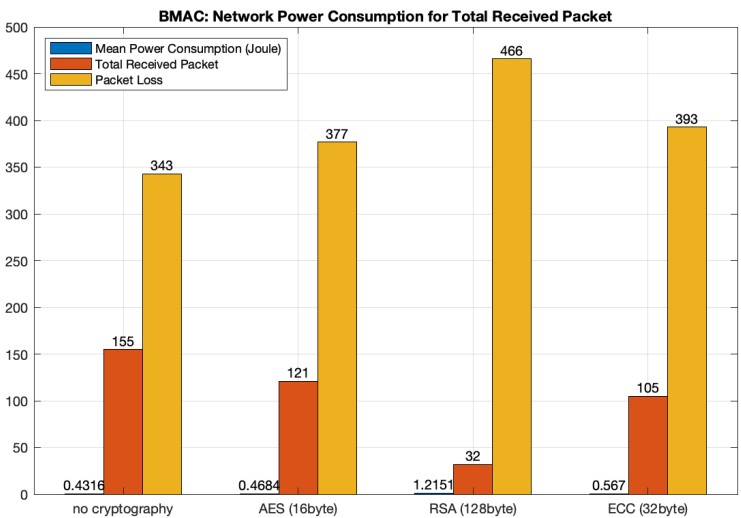

**Figure 13.** Average energy consumed, total number of received and loss packets for BMAC.

In particular, observing Figure 13 it is possible to make the following consideration for each cryptographic method:

Using AES

- The total received packet number is equal to 121 and there are 34 packets less than the initial setup described earlier.
- The energy consumption is equal to 0.46844 J and it is greater than the initial setup.

The total number of received packets is low since, considering the applied mitigation, the server is able to recognize tampered packets and then it discards them. Also, the packet size is greater (16 bytes) than the initial setup (10 bytes) due to the key length used for AES. Therefore, the energy consumption is strictly related to the size of the packet, and thus it is greater than the previous scenario.

Using RSA

- The total number of received packets in this case is equal to 32, so 123 packets less than the initial setup and 89 packets with respect to the scenario with AES. This is due to two factors: firstly, the ability of the server to discard the tampered and forged packets and, secondly, the low number of packets sent by sensor nodes since the packets with RSA have a greater length (128 bytes).
- The energy consumption is equal to 1.21514 J and it is greater than the initial setup, AES and even ECC. Since sensor nodes are equipped with limited power and computing resources, they are not able to handle packets which are characterized by a high size, and thus the network will experiment a bottleneck. As a consequence, a large number of packets gets lost.

Using ECC

- The total number of received packets is equal to 105, so 50 packets less than the initial setup, 16 less than AES and 73 less than RSA.
- The energy consumption is equal to 0.567043 J and it is greater than all other scenarios.

This data is due to the greater size of the packets, that in the case of ECC is equal to 256 bytes, according to key size used in ECC. This is with respect to the initial setup and with the AES scenario. Considering RSA, instead, the reason is due to the number of received packets being significantly higher, with 73 packets more added to the total number.

5.5.2. Impersonation Attack with LMAC

Considering Figure 14, without any mitigation and with the attacker sensor node, the total number of received packets is equal to 496, and the energy consumption is equal to 0.313092 J, considering a packet size of 10 bytes. The number of packets which get lost is 2. Including cryptographic techniques for mitigating the attack, the overall network performance is slightly different but the changes are not so evident as with the BMAC protocol. In particular, in each case (AES, RSA, and ECC) the total number of received packets is 396 with 102 packets lost. So, it is clear that with the LMAC no changes are highlighted, and for this reason the performance analysis will be focused only on the energy consumption.

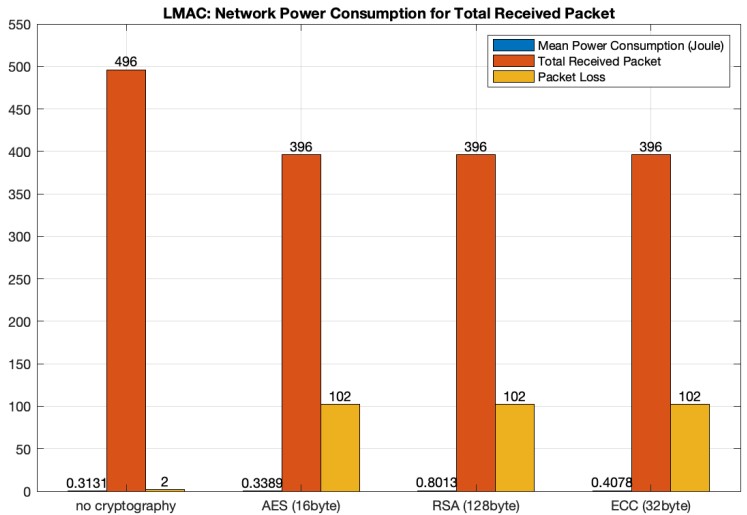

**Figure 14.** Average energy consumed, total number of received and loss packets for LMAC.

Using AES

- The energy consumption is equal to 0.338924 J and it is greater than the initial setup.

Using RSA

- The energy consumption is equal to 0.801263 J and it is greater than the initial setup and the setup with AES and ECC.

Using ECC

- Finally, in this setting, the energy consumption is equal to 0.407812 J, and therefore it is greater than AES and less than RSA.

RSA is the highest resource consuming technique, among all others, since it produces packets with a size of 128 bytes. Thus, also in this case, it is clear how the energy consumption is strictly related to the packets size.

*5.6. BMAC vs. LMAC: Overall Comparison*

In this section, an overall comparison of the performance showed and described in the previous sub-sections between LMAC and BMAC are provided. It shows the different cryptographic methods trying to determine, at the end, which is the best following a trade-off in terms of energy consumption and total number of received packets.

### 5.6.1. Bmac and LMAC Comparison Using AES

Considering the Figure 15 it is clear that LMAC, generally, is significantly better than BMAC since it is able to guarantee a higher number of received packets and a lesser number of packets that get lost as well as a lesser energy consumption.

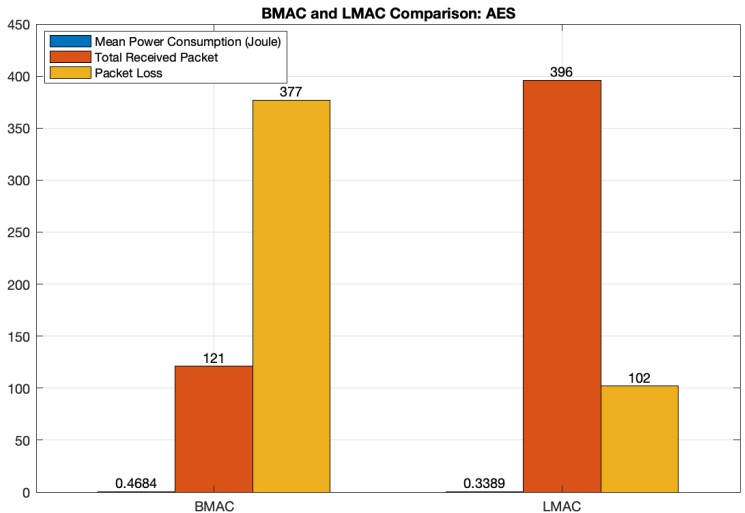

**Figure 15.** BMAC and LMAC comparison for AES: Average energy consumed, total number of received and loss packets.

### 5.6.2. Bmac and LMAC Comparison Using RSA

Considering the Figure 16, the number of received packets considering BMAC is less than the one with LMAC, and also it is registered a higher energy consumption. This result is due to the higher value given to the slot duration parameter in order to make sensors be able to handle packets with a size of 128 bytes. This leads, furthermore, sensors to send a small number of packets. Thus, also in this setting, LMAC is better than BMAC.

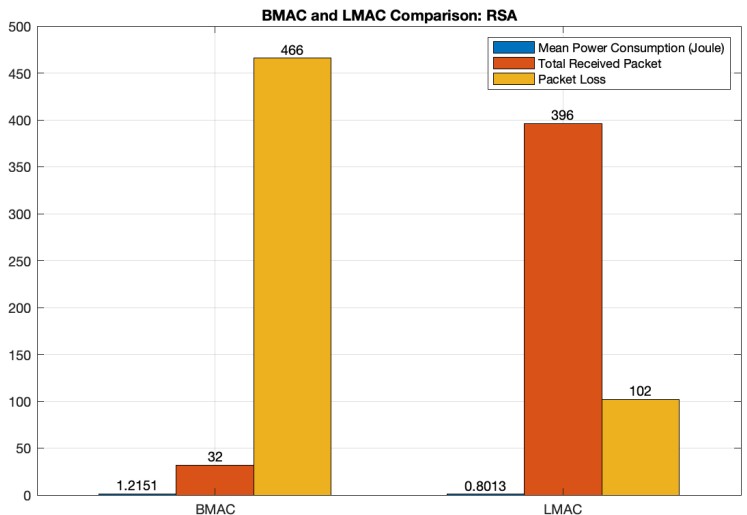

**Figure 16.** BMAC and LMAC comparison for RSA: Average energy consumed, total number of received and loss packets.

### 5.6.3. Bmac and LMAC Comparison Using ECC

Also in this setting, the experimental results show the same pattern observed in the previous setting, and LMAC is configured as the better choice. The experimental results are shown in Figure 17.

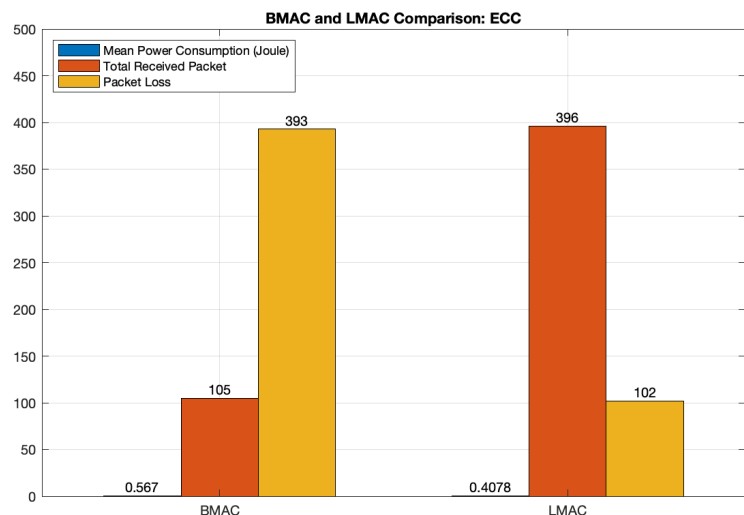

**Figure 17.** BMAC and LMAC comparison for ECC: Average energy consumed, total number of received and loss packets.

## 6. Conclusions

Mostly, WSNs' drawbacks are related to the limited resources (energy, computing, and bandwidth) of devices involved inside the network. Thus, WSNs could be easily taken as a target by malicious users and, due to the limited node resources, it is really hard to provide an adequate security mechanism. In this work, the security issues of two MAC protocols, BMAC and LMAC, are analyzed, tested, and compared. We have applied to these protocols different cryptographic techniques: AES, RSA, and ECC, in order to mitigate two specific attacks which could be carried out against these kind of networks leveraging the MAC protocol: the energy drain attack and the impersonation attack. We have evaluated the system under attack, and then we have applied mitigation techniques evaluating two output parameters: energy consumption and total number of received packets. Results show that security of exchanged data is guaranteed ensuring also confidentiality, integrity, and authentication, and that the LMAC protocol guarantees the best performance in comparison to the BMAC.

**Author Contributions:** Conceptualization, F.D.R. and M.T.; methodology, F.D.R.; software, M.G.S.; validation, A.F.G. and M.G.S.; data curation, M.T.; writing—original draft preparation, M.T., M.G.S. and A.F.G.; writing—review and editing, M.G.S. and M.T.; supervision, F.D.R. All authors have read and agreed to the published version of the manuscript.

**Funding:** This research received no external funding.

**Data Availability Statement:** Not applicable.

**Conflicts of Interest:** The authors declare no conflicts of interest.

## Abbreviations

The following abbreviations are used in this manuscript:

| | |
|---|---|
| AES | Advanced Encryption Standard |
| BMAC | Berkeley MAC |
| DES | Data Encryption Standard |
| DLP | Discrete Logarithmic Problem |
| DSA | Digital Signature Algorithm |
| ECC | Elliptic Curve Cryptography |
| ECDH | Elliptic Curve Diffie-Hellman |
| ECDSA | Elliptic Curve DSA |
| LMAC | Lightweight MAC |
| MAC | Medium Access Control |
| NIST | National Institute of Standards and Technology |
| RSA | Rivest Shamir Adelman |
| SMAC | Sensor MAC |

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
