# Peer review of "Security in Wireless Sensor Networks: A Cryptography Performance Analysis at MAC Layer"

_futureinternet, doi:10.3390/fi14050145_

Round 1

Reviewer 1 Report

Dear Authors,

The content of your article fits perfectly within the scope of Future Internet journal's "Security in Mobile Communications and Computing" Special Issue. One of the reasons for this is that a key research topic involves security issues in Wireless Sensor Networks (WSNs). This is very important for solutions that meet rapidly increasing requirements against a wide range of attacks and threats.

The Authors conducted a security analysis based on two MAC protocols using three cryptographic algorithms: AES, RSA and ECC. It is relevant and interesting.

The above-mentioned goal was based on the 25 publications analysed in the three initial sections of the article.

The experimental results, which were carried out on a suitable simulated WSN, are given and explained.

The Authors proved that the LMAC protocol provides the best solution in terms of energy consumption and guaranteed number of received packets in WSNs.

The paper contains some new data.

The paper is presented in logical way and overall written well.

The text is clear and easy to read.

The conclusion is consistent with the evidence and arguments presented and addresses the main question asked. The article makes reference to limitations.

Comments and Suggestions for Authors

  1. It would be best to clearly identify which of the previous works by all Authors constitute the foundation of the work presented in this article.
  2. The study considered the mitigation of two specific attacks: an energy-draining attack and an impersonation attack. What other alternatives were there?
  3. To what extent are the obtained results scalable, i.e. transferable to IPv6?
  4. The following typos are noticed in the article: please make the content of Figures 2-8 more readable.

Reviewer 2 Report

The authors in this article investigated WSNs security at the MAC layer. They started by discussing the key points of WSNs and then pointed out the WSNs security drawbacks. Performance evaluation experiments were conducted by exploring two MAC protocols using AES, RSA, and elliptic curve cryptography. The following are my comments on the manuscript:

  • Line 77: The authors mention “secrecy, integrity, and confidentiality”. What is the difference between secrecy and confidentiality? I suggest omitting secrecy and stick to “confidentiality” as a more formal well know definition. This needs to be done throughout the manuscript (i.e., replacing secrecy with confidentiality).
  • Line 81: “They can be grouped into three areas”. Is this an already existing or proposed taxonomy by the authors. I urge the authors to cite some attacks taxonomies that classify WSN attacks:
  1. Azzedin F, Albinali H. Security in Internet of Things: RPL Attacks Taxonomy. InThe 5th International Conference on Future Networks & Distributed Systems 2021 Dec 15 (pp. 820-825). https://doi.org/10.1145/3508072.3512286 (This taxonomy considers all RPL attacks including those inherited from WSN).
  2. Khanam S, Ahmedy IB, Idris MY, Jaward MH, Sabri AQ. A survey of security challenges, attacks taxonomy and advanced countermeasures in the internet of things. IEEE Access. 2020 Nov 11;8:219709-43.
  • The article contains English grammatical and composition mistakes. The article needs to be carefully proof-read. The followings are just some examples of such mistakes. It is the responsibility of the authors to thoroughly and carefully proof-read the entire article
  1. Line 93: “WSNs is an an hot and important topic”.
  2. Line 42-Line 45: The sentence spans multiple lines. This sentence needs to be rephrased.
  3. Line 54: Replace Wireless Sensor Networks with WSNs.
  • For the energy drain attack, it is not clear how is the sensor node forced to have send-interval value equal to the legacy nodes.

Round 2

Reviewer 1 Report

Dear Authors,

Thank you for clarifying all of the doubtful and blurred areas based on the comments made. I am fully satisfied with your answers. The article is now ready for publication.